# M-Carboxylic Acid Induced Formation of New Coordination Polymers for Efficient Photocatalytic Degradation of Ciprofloxacin

**DOI:** 10.3390/molecules27227731

**Published:** 2022-11-10

**Authors:** Jian Li, Xiaojia Wang, Yunyin Niu

**Affiliations:** 1Green Catalysis Center, and College of Chemistry, Zhengzhou University, Zhengzhou 450001, China; 2College of Ecology and Environment, Zhengzhou University, Zhengzhou 450001, China

**Keywords:** ciprofloxacin, coordination polymers, hydrothermal method, photocatalytic degradation

## Abstract

Four new 2–3D materials were designed and synthesized by hydrothermal methods, namely, {[(L1·Cu·2H_2_O) (4,4-bipy)_0.5_] (β-Mo_8_O_26_)_0.5_·H_2_O} **(1)**, {[(L1·Cu)_2_·(4,4-bipy)] (Mo_5_O_16_)} **(2)**, {Co(L1)_2_}_n_ **(3),** and {[(L1)_2_][β-Mo_8_O_26_]_0.5_·5H_2_O} **(4)**. [L1=5-(4-aminopyridine) isophthalic acid]. The degradation of ciprofloxacin (CIP) in water by compounds **1–4** was studied under visible light. The experimental results show that compounds **1–4** have obvious photocatalytic degradation effect on CIP. In addition, for compound **1**, the effects of temperature, pH, and adsorbent dosage on photocatalytic performance were also investigated. The stability of compound **1** was observed by a cycle experiment, indicating that there was no significant change after three cycles of CIP degradation.

## 1. Introduction

With the development of medical level, antibiotics have made great contributions to the prevention and treatment of diseases. Antibiotics can enter aquatic environments through various ways. The increase of their use and the water solubility, stability, and volatility made them present a “lasting” state in aquatic environments, resulting in a series of environments problems. Antibiotic wastewater is not facile to degrade, and its high biological toxicity makes it a ticklish problem in the field of sewage treatment. In recent years, high concentrations of antibiotics have been detected in many countries and regions.

Ciprofloxacin (CIP) antibiotics (Figure 1) are widely used in human and veterinary medicine because of their strong bactericidal ability, and lower toxicity and side effects [1,2]. However, the extensive use of CIP has also caused environmental pollution and posed a threat to the ecosystem and to human health. The removal of CIP from environment has become a mandatory issue already [3,4]. Unfortunately, the conventional chemical and physical methods lack enough efficiency for its removal and the new biochemical treatment method often produces byproducts that are more harmful.

Nowadays, the migration behavior and degradation mode of antibiotics in aquatic environments have become a research hotspot. Photochemical degradation is an important way for the migration and transformation of antibiotics in aquatic environments [5], which has a significant impact on the environmental toxicological effects of these substances. In recent years, photocatalytic technology has been widely used in water pollution control, which shows the important application prospect and potential of photocatalytic technology [6,7]. The photocatalytic degradation processes provided an ideal technique for the transportation and degradation of CIP [8,9,10]. Compared with the methods mentioned above, using photocatalytic technology can degrade CIP efficiently without secondary pollution.

As a new type of functional molecular materials, coordination polymers are formed by the self-assembly process of organic ligands and metal ions. In the relevant research literature [11,12,13,14,15,16], they are also called porous coordination polymers (PCP) or organic-inorganic hybrid materials. Coordination polymers materials developed rapidly and became a research hotspot in the fields of chemistry [5], environment [17], materials [18], etc.. The diversity of metal ions and organic ligands, as well as the different coordination modes between them [13,19], determine the structural diversity of these materials. The synthetic fibers of coordination polymers, synthetic methods, the specific configuration formed by the inherent ligands between organic ligands and metal ions lead to the diversity of the functions. The unique structure and function also show broad application prospects in the selective adsorption and catalytic degradation of toxic and harmful substances [20]. Due to the advantages of high specific surface area, self-assembled structures, and evenly distributed active sites, coordination polymer materials have great potential in the fields of catalyst preparation, adsorption, gas storage, and photocatalytic degradation [11,12,14,16,21]. In the published articles on the photocatalytic degradation of ciprofloxacin, most of the catalysts are doped or loaded modified materials [8,9,10]. Therefore, we imagine whether materials with excellent degradation performance can be directly obtained through a simple one-step reaction.

We have been working on the exploration of organic-inorganic hybrid materials for a long time [5,13,22,23]. Carboxylic acid ligands usually have good metal ion binding ability. Carboxylate is a hard base, which can form strong coordination bonds with various common metal ions. Moreover, carboxylate has negative charges, which can neutralize the positive charges of metal ions and metal clusters, so that the pores of porous complexes do not need to contain counter anions, which is conducive to improving the stability of the structure. Based on this, we chose the unreported ligand 5-(4-aminopyridine) isophthalic acid (Appendix A) to react with metal salts by hydrothermal method and obtained four new novel organic-inorganic hybrid materials **1–4**. It is worth mentioning that compounds **1–4** are directly obtained by hydrothermal reaction, without further doping and modification. Compounds **1–4** were systematically characterized by infrared spectroscopy (IR), elemental analysis, and powder X-ray diffraction (PXRD). Under simulated light, compounds **1–4** have a good degradation effect on CIP and show a good photocatalytic performance. In addition, the effects of temperature, pH, and catalyst dosage on the degradation of CIP by compound **1** were investigated. Cyclic experiments show the workability of the photocatalyst.

## 2. Results

### 2.1. Crystal Structure

X-ray single-crystal structural analysis indicated that compound **1** crystallized in triclinic system (space group P-1). As shown in Figure 2a, its asymmetric structural unit is composed of an L1 ligand, a Cu (II) ion, half of a 4,4-bipyridine molecule, a terminal coordination water molecule and a μ- coordination water molecule, half a [β-Mo_8_O_26_]^4−^. Cu coordinate to a O atom in the one carboxylate of L1 ligand, N atom in the 4,4-bipyridine molecule, a terminal O and two μ-O atoms in the coordination water molecule, and an O in [β-Mo_8_O_26_]^4−^, forming an octahedral hexacoordinate mode. The bond length around the Cu ion ranges from 1.912 (6)–2.649 (6) Å (Cu1-O) and 1.972 (7) Å (Cu1-N), the bond angel around Cu ion ranges from N-Cu-O = 45.947 (12)–92.47 (12)°, and O-Cu-O = 44.585 (84)–60.224 (88)°. L1 ligand acts as a terminal ligand to coordinate with Cu atom. Via the coordination and connected mode with organic and polymolybdate ligands, these Cu_2_(H_2_O)_4_ dimer can be infinitely extended in the space to form a 3D network structure. Figure 2b is the stacking diagram of **1**.

X-ray single-crystal structural analysis indicated that compound **2** crystallized in monoclinic system with space group I2/C. As shown in Figure 2c, the smallest structural unit of compound **2** is composed of two Cu (II) ions, two L1 ligands, a 4,4-bipyridine molecule, and a [Mo_5_O_16_]^2−^ anion cluster. Both Cu (II) ions present a hexacoordinated octahedral configuration formed by the coordination with two O atoms from two L1 ligands, three O atoms from three [Mo_5_O_16_]^2−^ anion clusters, and a N atom from a 4,4-bipyridine molecule. Interestingly the two carboxylate groups in L1 ligand act as different bridging modes; one links a Cu and a Mo, while another links two Cu atoms and a Mo atom. Via the coordination and connected mode with organic and polymolybdate ligands, these Cu_2_(H_2_O)_4_ dimers can be infinitely extended in the space to form a 3D network structure. Figure 2d is the stacking diagram of **2**.

Compound **3** belongs to the orthorhombic system and pbcn space group. As shown in Figure 2e, the asymmetric structural unit of compound **3** is composed of one Co (II) ion and two L1 ligands. In compound **3**, each Co is tetra-coordinated by coordinating with four O atoms from four different L1 ligands. The main bond length and bond angle around the Co (II) atom are Co1-O2 = 1.954 (5) Å, Co1-O3 = 1.946 (5) Å, O2-Co1-O3 = 40.323 (141)°, O3-Co1-O3 = 107.5 (3)°, respectively. L1 ligand acts as a bridging ligand to link two different Co centers. Via the coordination and connected mode, compound **3** forms a 1D chain structure. Figure 2f is the stacking diagram of **3**.

X-ray single-crystal structural analysis indicated that compound **4** crystallized in triclinic system, P-1 space group. The asymmetric structural unit of compound **4** consists of half [β-Mo_8_O_26_]^4−^, two L1 ligands, and five free H_2_O molecules (Figure 2g). Figure 2h is the stacking diagram of compound **4**, where it can be seen that [β-Mo_8_O_26_]^4−^ anion clusters are filled among the organic cationic ligands. There is not coordination bond between the organic cationic ligand L1 and the [β-Mo_8_O_26_]^4-^ anion cluster; they form a 2D supramolecular structure through an electrostatic interaction, intermolecular force, and hydrogen bond (C-H ··· O).

Compounds **1–4** are all obtained by hydrothermal synthesis, but different coordination rules are shown between ligand L1 and metals. In compound **1**, one carboxylate on ligand L1 is coordinated with Cu, while in compound **2**, one carboxylate is coordinated with Cu, and the other carboxylate is coordinated with Mo. In compound **3**, Co is coordinated by two carboxylate groups. In compound **4,** there is no coordination contact between ligand L1 and Mo, but a charge balanced supramolecular compound is formed.

### 2.2. XRD Analysis

The PXRD pattern of compounds **1–4** were recorded and compared with the simulated single-crystal diffraction data in order to affirm the purity of the compounds. For compounds **1–4,** the position of the peaks are basically consistent with the simulated patterns generated from the results of the single crystal diffraction data, indicating the purity of products (Figure 3a–d). The difference in reflection intensities between the simulated and experimental patterns was due to the variation in the preferred orientation of the powder sample during the collection of the experimental PXRD data. Crystal data for compounds **1–4** were summarized in detail in Appendix A. Selected bond lengths and bond angles were put in Appendix A.

### 2.3. TG Analysis

In order to investigate the thermal stability of compounds **1–4**, TG analysis was performed. As shown in Figure 4, compounds **1–4** remained substantially unchanged from room temperature to 300 °C. The pyrolysis process of compounds **1**, **2,** and **4** are very similar, probably because of the similar ligand constitution containing L1 and molybdate. Compared with other compounds, the pyrolysis process of compound **3** is quite different, which may be because there is no molybdic acid in its structural composition.

### 2.4. Band Gap Analysis

As shown in Figure 5, the band gap values of compounds **1–4** are 2.12 eV, 1.89 eV, 2.01 eV, and 2.27 eV, respectively. This shows that compounds **1–4** are expected to be semiconductors when exposed to visible light and have potential photocatalytic activity. Bandgap is an important characteristic parameter of semiconductors. Its size is related to the crystal structure and the bonding properties of atoms. The diffuse reflectance UV-Vis spectra of Compounds **1-4** can be seen in Appendix A. The different bandgap values of compounds **1–4** may be caused by their different crystal structure and the bonding properties. Based on the size of band gap and the reported literature [13], we speculated that the compounds **1–4** might have potential photocatalytic activity, so we carried out subsequent experimental exploration.

### 2.5. Photocatalytic Activity

Ciprofloxacin (CIP) antibiotics are widely used in human and veterinary medicine because of their strong bactericidal ability, lower toxicity, and side effects [24,25]. However, this also led to an increase in the concentration of CIP antibiotics in aquatic environments, resulting in a series of aquatic environments pollution. Therefore, CIP was used as the target degradation product for photocatalytic degradation experiment to explore the photocatalytic performance of compounds **1–4**. Take compound **1** as an example. First, the initial absorbance of CIP was surveyed in the range of 300–450 nm. Then, 0.94 mol% of compound **1** was added to an aqueous solution of CIP. Before lighting, the mixture was magnetically stirred in the dark for 30 min to achieve the adsorption equilibrium between compound **1** and the CIP solution. A 500 W mercury lamp was chosen as a visible light source, and during the photocatalytic reaction process, 3 mL of the suspension was taken out of the mixed solution at regular intervals, and the supernatant was analyzed on a UV-Vis spectrophotometer after centrifugation. The photocatalytic activity of compounds **1–4** were measured by degrading the aqueous solution of CIP under visible light. In the experiment without a catalyst, no significant degradation of CIP was observed, but CIP began to degrade after the addition of compounds, indicating that the effect of light on CIP degradation is negligible, and compounds **1–4** can be used as photocatalysts for CIP. For the reusability test, the supernatant was poured out after the degradation reaction is complete, and fresh CIP solution (20 mL, 25 mg/L) was added to the mixture. Subsequently, the photocatalytic reaction was continuously, magnetically stirred under the irradiation of a 500 W high-pressure xenon lamp. This operation was repeated three times [13].

The calculation formulas of degradation efficiency [26] and removal rate are as follows: C_0_ is the CIP concentration when the illumination time is 0, and C_t_ is the CIP concentration when the illumination time is t. A_0_ and A_t_ are the absorbance of CIP when the illumination time is 0 and t, respectively. It can be seen from the Figure 6a that CIP is hardly degraded under visible light without catalyst. After adding compounds **1–4,** respectively, the degradation rates reached 86.95%, 67.18%, 62.02%, and 59.34%, respectively. This showed that compounds **1–4** have a degrading effect on CIP solution under visible light. It can be seen from Figure 6b that the reaction rate constants of ciprofloxacin degradation by compounds **1–4** are K = 0.00444 min^−1^, K = 0.00373 min^−1^, K = 0.00391 min^−1^ and K = 0.00382 min^−1^ respectively, indicating that compound **1** has a faster degradation rate of CIP than compounds **2**, **3,** and **4**. We found that the degradation effect of compound **1** obtained by our one-step reaction is comparable with the reported modified Bi_2_Ti_2_O_7_/TiO_2_/RGO composite [8]. Based on this, in the follow-up research work, we used compound **1** as a catalyst to study the effects of pH, temperature, and catalyst amount on the catalytic degradation of CIP.

The pH value of the solution is an important factor affecting the photocatalytic performance [27,28,29]. The photocatalytic degradation of CIP solution by compound **1** at different pH was studied. Before the formal experiment, adjust the pH with nitric acid and sodium hydroxide to prepare CIP solutions with pH values of 3, 5, 7, and 9, respectively. It can be seen from the Figure 7a that when the pH value is lower than 7, the degradation rate of CIP by compound **1** increased with the increase of pH value. When the pH values were 5 and 7, the final degradation efficiency reached 64.47% and 69.88%, respectively. It can also be seen from Figure 7b that when the pH value is 7, the degradation rate constant (k = 0.00444 min^−1^) of CIP by compound **1** is the largest. The results showed that the neutral condition was more suitable for the degradation of CIP by compound **1**.

Temperature also has a certain influence on the catalytic degradation effect of the catalyst [30,31,32]. Based on this, in this study, we explored the degradation effect of compound **1** on CIP at different temperatures. It can be seen from the Figure 8a that the degradation effect increased first and then decreased with the increase of temperature. At different temperatures, the final degradation rates of CIP reached 69.88%, 78.9%, and 71.25%, respectively. The results showed that the degradation effect of compound **1** on CIP increased and decreased slightly with the increase of temperature, so the effect of temperature on the photocatalytic degradation of CIP by compound **1** was not significant. The rate constant of k = 0.00444 min^−1^ under 30 °C, k = 0.00574 min^−1^ under 40 °C and k = 0.00446 min^−1^ under 50 °C, the degradation rate of CIP by compound **1** was slightly higher at 40 °C.

In the degradation experiment, we also explored the influence of the amount of catalyst on the catalytic effect. CIP solution was placed under constant stirring conditions, and the degradation of CIP was studied by adding different doses of compound **1**. It can be seen from the Figure 9a that in the CIP/compound **1** system, when the amount of catalyst is from 5 mg to 20 mg, the degradation efficiency of CIP first increases and then decreases. The reason for this state is that the aggregation of excess compound **1** particles hinder and inhibit the scattering and transmission of light in the solution, while the organics adsorbed on the photocatalyst will reduce the utilization of light [33,34,35]. After adding different doses of compound **1**, the final degradation rates reached 81.13%, 86.95%, and 69.97%, respectively. It can also be seen from Figure 9b that among the three doses, when the amount of compound **1** is 10 mg, the degradation rate of CIP is the highest.

To investigate the practical value of compound **1** in photocatalytic degradation of CIP, we explored the regeneration and stability of compound **1**. The photocatalytic stability of compound **1** is shown in Figure 10. After the photocatalytic degradation of CIP solution finished, compound **1** was collected. The surface was cleaned with deionized water to remove the residual CIP in the previous experiment, and then put the collected compound **1** into the fresh CIP solution to start a new cycle. The stability and reusability of compound **1** for degradation of CIP was investigated by three consecutive cycles. In the cycle experiment, all experimental conditions were exactly the same as the first experiment. As shown in Figure 10, the degradation efficiency of CIP has not decreased significantly. After three cycles, the degradation efficiency can still reach 80.47%. The results showed that compound **1** can be used as a stable photocatalyst for the photocatalytic degradation of CIP. The stability of compound **1** before and after the photocatalytic reaction was further verified by scanning electron microscope analysis, and the results are shown in Figure 11.

To further investigate the internal mechanism behind compound **1** for CIP degradation, active substance capture experiments were performed. In order to clarify the active substances produced by the catalyst in the catalytic degradation of CIP, different free radical scavengers were added to the photocatalytic reaction system under the same light conditions. Specifically, three active substances of EDTA-2Na (capture h^+^), BQ (capture ^·^O_2_^−^) and IPA (capture ^·^OH) are mainly used in the photocatalytic process [36,37]. Figure 12 shows that after adding BQ and IPA to the above solution, the degradation rates of CIP by BQ and IPA are 45.38% and 31.45%, respectively. When EDTA-2Na is added to the reaction system, it can greatly inhibit the photocatalytic degradation of CIP, which indicated that h+ plays an important role. Although the addition of BQ interfered with photocatalytic activity, the photocatalytic degradation rate only decreased to 45.38%, indicating that ·OH is not a critical reactant.

## 3. Conclusions

In summary, four new compounds were synthesized by hydrothermal method. They were characterized by single-crystal X-ray diffraction, elemental analysis, IR, powder X-ray diffraction and TG analysis, they have good catalytic potential for photodegradating CIP. Among them, compound **1** has a faster degradation rate of CIP. The stability of compound **1** was observed by a cycle experiment, indicating that there was no significant change after three cycles of CIP degradation. In subsequent studies, in order to make our research more comprehensive, it is sensible to perform a hot test to check heterogeneity of the reaction. Moreover, it is expected to explore the size and shape effects on the photocatalytic property.

## 4. Materials and Methods

### 4.1. Materials

The ligand L1 was synthesized (Appendix A) similarly to the reference method [33]. All other reagents for the synthesis and analysis were commercially available and used without further treatment.

### 4.2. Methods

#### 4.2.1. Synthesis Methods

##### Synthesis of Compound 1

A solution of L1 (0.0086 g, 0.025 mmol), (NH_4_)_6_Mo_7_O_24_·4H_2_O (0.0309 g, 0.025 mmol), CuCl_2_·2H_2_O (0.0043 g, 0.025 mmol), 4,4-bipyridine (0.0038 g, 0.025 mmol), and H_2_O (10 mL) was stirred under ambient conditions, and the pH was adjusted to 5 with HCl (2 mol/L), then sealed in a Teflonlined steel autoclave, heated at 120 °C for 3 days, and cooled to room temperature. The resulting product was recovered by filtration, washed with distilled water, and dried in air. Yield: 45%. IR (KBr, cm^−1^): 3361.96 (m), 3224.37 (m), 1875.07 (w), 1698.07 (m), 1651.87 (s), 1111.22 (w), 1035.15 (m), 945.63 (m), 919.62 (m), 668.59 (m), 506.36 (m), 471.46 (m). Elemental Anal. Calc. for C_19_H_23_CuMo_4_N_3_O_20_ (772.54): C, 29.54; H, 2.98; N, 5.44. Found: C, 29.57; H, 3.01; N, 5.47.

A similar procedure was followed to prepare compounds **2**–**4** [see the Appendix A].

#### 4.2.2. Characterization Methods

The infrared spectrum was measured on Shimazu IR 435 spectrometer, in the form of a KBr disk (4000–400 cm^−1^). Elemental analyses (C, H, and N) were performed on a FLASH EA 1112 elemental analyzer. Amodel NETZSCHTG209 thermal analyzer was used to record simultaneous TG curves in flowing air atmosphere of 20 mL·min^−1^ at a heating rate of 5 °C·min^−1^ in the temperature range 25–800 °C using platinum crucibles. Suitable single crystals of **1–4** were carefully selected under an optical microscope and glued to thin glass fibers. The crystallographic data of compounds are acquired on a Bruker APEX-II area detector diffractometer equipped with graphite monochromatic Cu-Kα radiation (λ = 1.54184 Å) at 293(2) K. The structure was refined with full-matrix least-squares techniques on F^2^ using the OLEX2 program package. The CCDC reference numbers are 2,085,864, 2,124,165, 2,085,863, and 2,085,866 for compounds **1**, **2**, **3**, and **4** respectively.

## Figures and Tables

**Figure 1 molecules-27-07731-f001:**
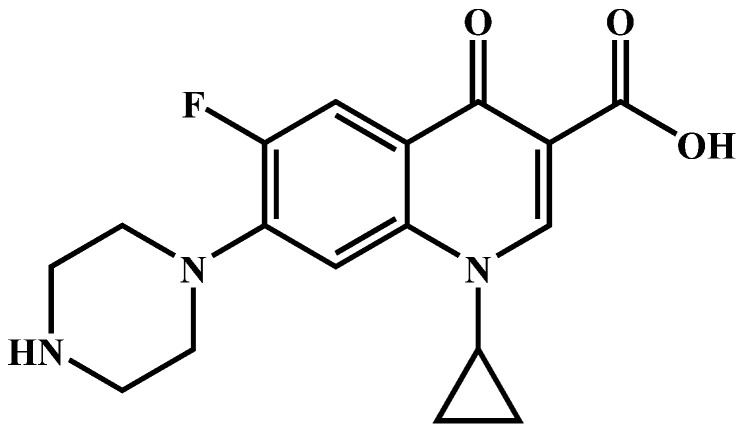
The atomic structure of CIP.

**Figure 2 molecules-27-07731-f002:**
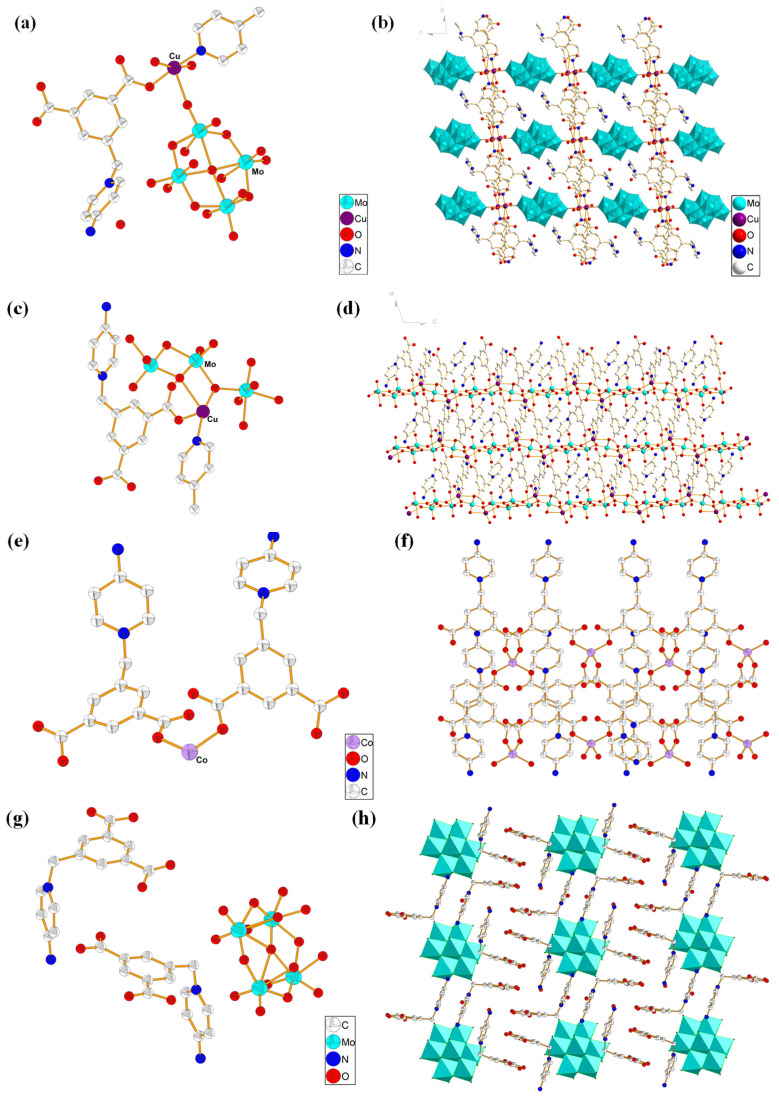
(**a**) Monomer structure diagram of compound **1** (with the H atom omitted), (**b**) Stacking diagram of compound **1**, (**c**) Structural unit diagram of compound **2**, (**d**) Stacking diagram of compound **2**, (**e**) Asymmetric structural unit diagram of compound **3**, (**f**) Stacking diagram of compound **3**, (**g**) Asymmetric structural unit diagram of compound **4**, (**h**) Stacking diagram of compound **4**.

**Figure 3 molecules-27-07731-f003:**
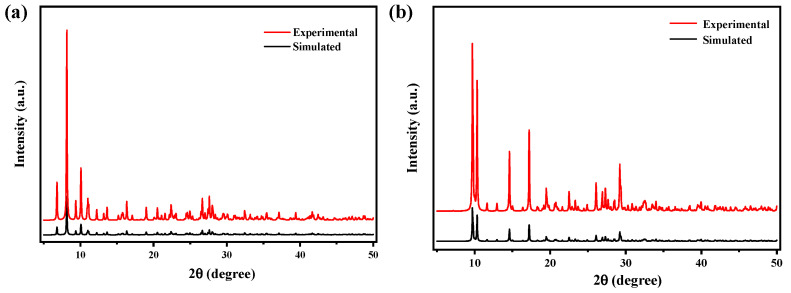
(**a**) PXRD pattern of compound **1**, (**b**) PXRD pattern of compound **2**, (**c**) PXRD pattern of compound **3**, (**d**) PXRD pattern of compound **4**.

**Figure 4 molecules-27-07731-f004:**
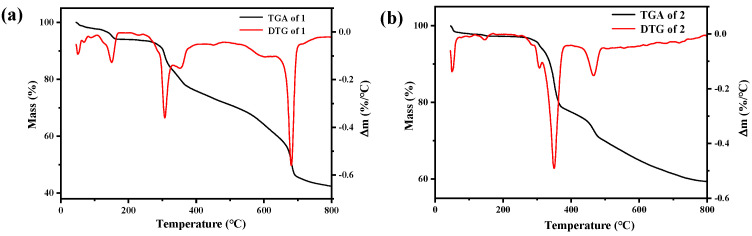
The TG and DTG curve of compounds **1(a)**, **2(b)**, **3(c)**, and **4(d)**.

**Figure 5 molecules-27-07731-f005:**
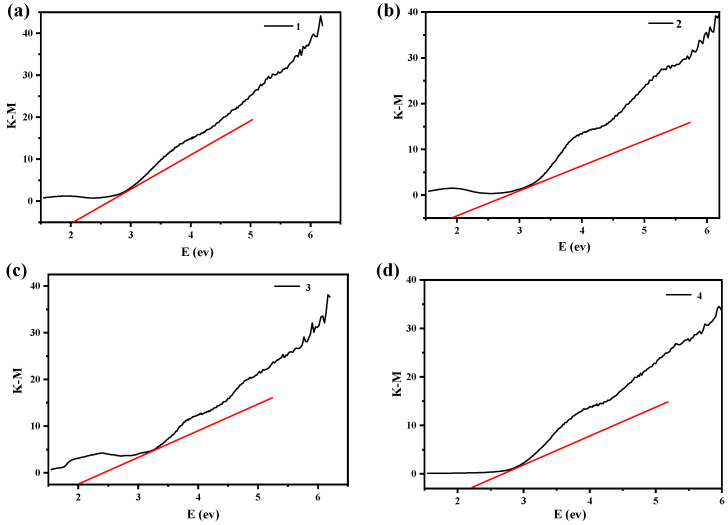
Optical band gap of compounds **1(a)**, **2(b)**, **3(c)**, and **4(d)**.

**Figure 6 molecules-27-07731-f006:**
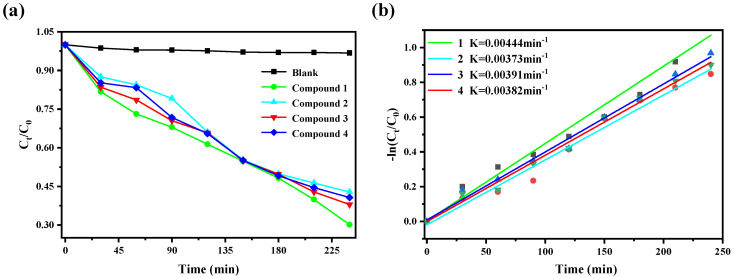
(**a**) The catalytic efficiency of different catalysts on CIP, (**b**) Quasi−first order kinetics of ciprofloxacin degradation by different catalysts.

**Figure 7 molecules-27-07731-f007:**
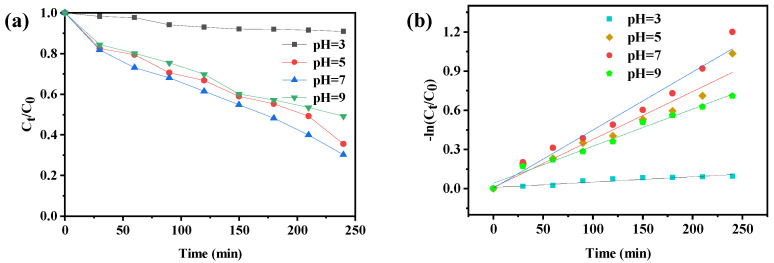
(**a**) Degradation rate of compound **1** to ciprofloxacin under different pH conditions, (**b**) Quasi-first order kinetic reaction rate at different pH values.

**Figure 8 molecules-27-07731-f008:**
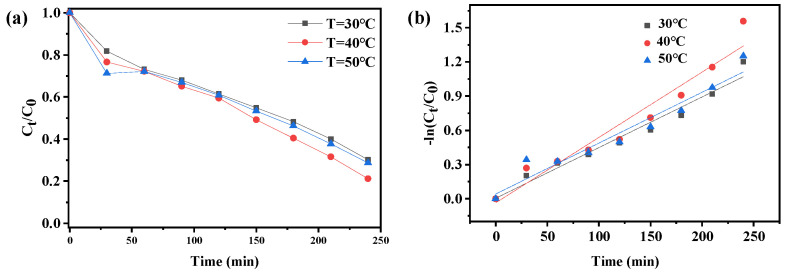
(**a**) Degradation rate of compound **1** to ciprofloxacin under different temperatures, (**b**) Quasi−first order kinetic reaction rate at different temperatures.

**Figure 9 molecules-27-07731-f009:**
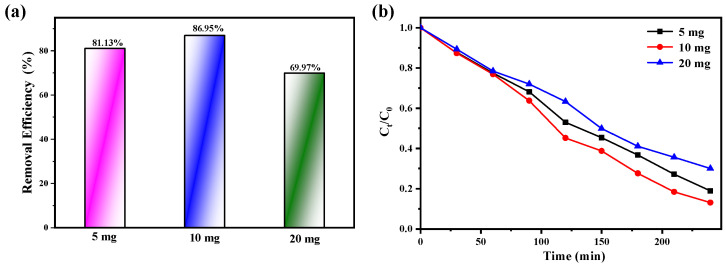
(**a**) Effect of catalyst **1** dosage, (**b**) Degradation rate of compound **1** to ciprofloxacin under different doses.

**Figure 10 molecules-27-07731-f010:**
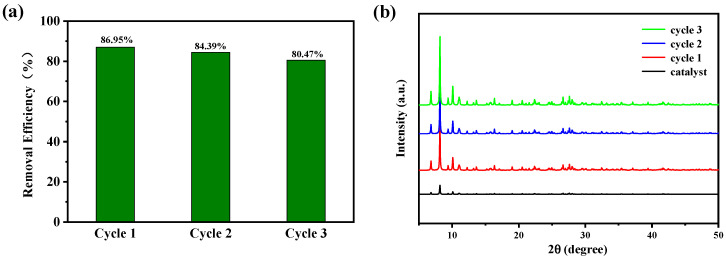
(**a**) Cycle test of compound **1** to degrade CIP at 30 °C, (**b**) PXRD patterns of compound **1** before and after the catalytic reaction.

**Figure 11 molecules-27-07731-f011:**
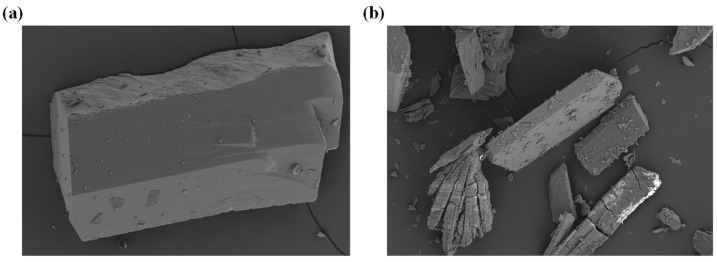
The SEM crystal diagram of compound **1** before (**a**) and after (**b**) the catalytic reaction.

**Figure 12 molecules-27-07731-f012:**
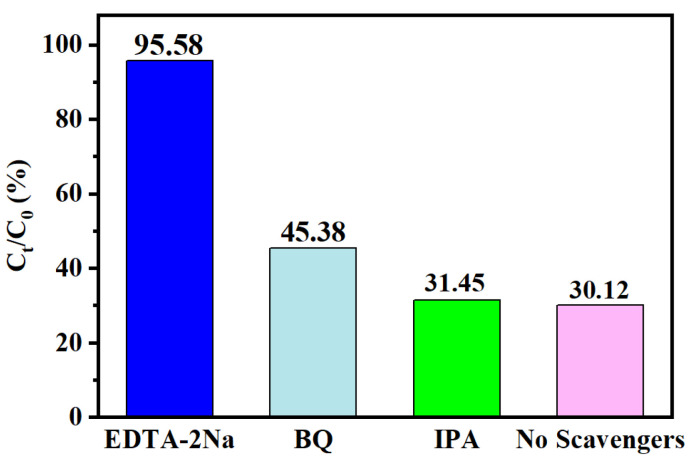
Trapping experiment of active species during the photocatalytic degradation of CIP over catalyst **1** under visible light.

## Data Availability

All data generated or analyzed during this study are included in this article.

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
