# Peer review of "M-Carboxylic Acid Induced Formation of New Coordination Polymers for Efficient Photocatalytic Degradation of Ciprofloxacin"

_molecules, 2022, doi:10.3390/molecules27227731_

Round 1

Reviewer 1 Report

Recommendation: The manuscript may be publishable after a major revision

Yunyin Niu and et al. report on photocatalytic activity of coordination polymers based on 5-(4-aminopyridine-1-yl) isophthalic acid usingdecomposition of ciprofloxacin as a model reaction. Four coordination polymers were characterized by single crystal X-ray difraction and powder X-ray analysis was performed to prove the homogeneity of the materials synthesized under hydrothermal conditions. Thus, the materials are correctly characterized. Then their efficiency as photocatalysts was investigated. Unfortunately, I didn’t find the Supporting Information cited in the main text (Supplementary Materials: The supporting information includes: Synthesis of Compounds 2-4; 294 Crystal data for compounds 1-4 (Table S1). Selected bond lengths and bond angles (Table S2)).

I recommend this manuscript to publication only after corrections of following points

1.     The experimental PXRD pattern of compound 4 and  the PXRD pattern simulated from single-crystal X-ray diffraction data are rather different. Please, explain this difference in the text.

2.     Page 1, line 29 “Ciprofloxacin (CIP) antibiotics (Fig. 1) are widely used in human and veterinary  medicine because of their strong bactericidal ability, less toxic and side effects.” Please add the references.

3.     Page 1 line 32 “The removal of CIP from the environment has become a mandatory issue already .“ Please add the references.

4.     Page 2, line 50. “they are also called porous coordination polymers (PCP) or organic-inorganic hybrid materials.“ Please precise the difference between MOFs, PCP and organic-inorganic hybrid materials.

5.     Page 2, line 61 “especially in photocatalytic degradation[7-10, 12].“  Please explain special role of MOFs in photocatalytic degradation as compared to other organic-inorganic hybrid materials or change this discussion.

6.     Page 2, line 76. “This study provided a theoretical basis and reference for the photocatalytic degradation of antibiotic pollutants in water.” Please, prove the novelty of your approach, add this statement in the conclusions or delete this sentence.

7.     Page 3, line 91 “Compound 1 formed a 3D structure through above coordination bond and hydrogen bonding interaction (Figure 2b).” The description of structure is incomplete. Please, precise the dtype of coordination polymer (1D,2D or 3D) and the role of hydrogen bonding in the stabilization of this polymer.

8.     Structural part: The discussion of structures 1-4 is too brief. Please, add at least the description of coordination mode of ligand (which donor centers are coordinated and which are free) for all compounds. Add the description of coordination framework (1D, 2D or 3D) for each compound. Discuss porosity of this polymers. The stacking diagrams which are shown in figure 2 have to be revisited since they are unclear.

9.     Page 6 line 165 “Then, 10 mg of compounds 1-4 were added to an  aqueous solution of CIP”. This information doesn’t allow the estimation of photocatalyst amount introduced into the reaction. Please, give this amount in mol% (use the brutto formula for each material to perform these calculations).

10.  Figure 6. As shown in this figure, the photocatalytic efficiency of four coordination polymers is comparable despite these materials are based on different metal nodes. Please, explain this results.

11. Figure 9. According this figure, an increase of catalyst loading above 10 mg does not allow to increase the reaction rate. The authors should continue these studies to show the optimal amount of catalyst. Please, complete these studies using 1, 3 and 5 mg of the catalyst, for example.

12. Heterogeneity of the reaction should be checked. Please, performed hot test.

13. Please, compare the efficiency of your catalyst to that of classical photocatalyst Degussa P25 titanium dioxide, for example.

11. Elemental analyses do not correspond to propose structures. For example: for compound 1: Elemental Anal. Calc. for C19H23CuMo4N3O20 (772.54): C, 29.54; H, 290 2.98; N, 5.44. Found: C, 30.99; H, 3.01; N, 5.47.

The difference between C teor and Cexperm. is large than 0.5. Please, add a comment or correct brutto formula of the material.

There are also minor points to correct:

1.The title of article is very long but contains only one keyword (ciprofloxacin) that is inconvenient for data bases. Please, change the title focusing on its research content.

2. The Figure 1 is missing in my version of the article.

3. Page 2, line 65 “a long time[18-20]” : Please add a point after [18-20].

4. Page 3, line 102 “a porous 3D cation  layer (Figure 2d) ”. Please change this sentence. A layer possesses 2D structure and cannot be 3D.

5. Page 3, line 106 “forms a tetra coordination mode “. Please use standard way to describe coordination shell of metal ions.

6. Figure 2a and b. Use the same color code for Mo and Cu atoms.

7. Page 6, line 186. Please, use bold for labels 1-4.

8.Please add points in journals abbreviation.

9.Ref. 33. Please, delete [J]. and replace 13: by 13, (replace semicolon by comma)

Author Response

                 Responses to the Reviewer # 1 s’ comments

Comments:Yunyin Niu and et al. report on photocatalytic activity of coordination polymers based on 5-(4- aminopyridine-1-yl) isophthalic acid usingdecomposition of ciprofloxacin as a model reaction. Four coordination polymers were characterized by single crystal X-ray difraction and powder X-ray analysis was performed to prove the homogeneity of the materials synthesized under hydrothermal conditions. Thus, the materials are correctly characterized. Then their efficiency as photocatalysts was investigated. Unfortunately, I didn’t find the Supporting Information cited in the main text (Supplementary Materials: The supporting information includes: Synthesis of Compounds 2-4; 294 Crystal data for compounds 1-4 (Table S1). Selected bond lengths and bond angles (Table S2)). I recommend this manuscript to publication only after corrections of following points

Reply: Sincerely thank you for your valuable comments on our manuscript. We have revised the manuscript according to your comments carefully. The specific modifications are listed below.

Comment 1: “the Supporting Information cited in the main text”

Reply: We have marked the relevant parts of the main text that need to cite the support information. There are three marks in 2.2, 4.1 and 4.2.1 respectively.

Comment 2: The experimental PXRD pattern of compound 4 and the PXRD pattern simulated from single-crystal X-ray diffraction data are rather different. Please, explain this difference in the text.

Reply: Thanks for your comments. For compound 4 the difference in reflection intensities between the simulated and experimental patterns was due to the variation in the preferred orientation of the powder sample during collection of the experimental PXRD data. This explanation has been added to the part 2.2. XRD Analysis.

Comment 3: Page 1, line 29 “Ciprofloxacin (CIP) antibiotics (Fig. 1) are widely used in human and veterinary medicine because of their strong bactericidal ability, less toxic and side effects.” Please add the references.

Reply: We have added references here.

Comment 4: Page 1 line 32 “The removal of CIP from the environment has become a mandatory issue already.” Please add the references.

Reply: We have added references here.

Comment 5: Page 2, line 50. “they are also called porous coordination polymers (PCP) or organic-inorganic hybrid materials.” Please precise the difference between MOFs, PCP and organic-inorganic hybrid materials.

Reply: PCP, also known as MOF, are crystalline materials composed of metal ions and organic-inorganic ligands connected by coordination bonds. They all belong to organic-inorganic hybrid materials. Organic-inorganic hybrid materials are a larger category, including supramolecular compounds formed by non covalent interactions between ligands and metals.

Comment 6: Page 2, line 61 “especially in photocatalytic degradation [7-10, 12].” Please explain special role of MOFs in photocatalytic degradation as compared to other organic-inorganic hybrid materials or change this discussion.

Reply: We have modified the description in relevant positions of the manuscript.

Comment 7: Page 2, line 76. “This study provided a theoretical basis and reference for the photocatalytic degradation of antibiotic pollutants in water.” Please, prove the novelty of your approach, add this statement in the conclusions or delete this sentence.

Reply: This sentence was deleted in the original position of the manuscript.

Comment 8: Page 3, line 91 “Compound 1 formed a 3D structure through above coordination bond and hydrogen bonding interaction (Figure 2b).” The description of structure is incomplete. Please, precise the dtype of coordination polymer (1D,2D or 3D) and the role of hydrogen bonding in the stabilization of this polymer.

Reply: We deleted some inappropriate descriptions and tried to describe the crystal structure of compound 1 in more detail. The dtype of coordination polymer (1D,2D or 3D) and the role of hydrogen bonding in the stabilization of this polymer are precisely described.

Comment 9: Structural part: The discussion of structures 1-4 is too brief. Please, add at least the description of coordination mode of ligand (which donor centers are coordinated and which are free) for all compounds. Add the description of coordination framework (1D, 2D or 3D) for each compound. Discuss porosity of this polymers. The stacking diagrams which are shown in figure 2 have to be revisited since they are unclear.

Reply: We added the description of crystal structure. Since the porosity of these compounds is low, we did not make such discussion in the manuscript. In order to make the image clearer, we redraw the crystal structure diagram in Figure 2.

Comment 10: Page 6 line 165 “Then, 10 mg of compounds 1-4 were added to an aqueous solution of CIP”. This information doesn’t allow the estimation of photocatalyst amount introduced into the reaction. Please, give this amount in mol% (use the brutto formula for each material to perform these calculations).

Reply: We have revised it in the original manuscript according to your requirements.

Comment 11: Figure 6. As shown in this figure, the photocatalytic efficiency of four coordination polymers is comparable despite these materials are based on different metal nodes. Please, explain this result.

Reply: The photocatalytic efficiency of four coordination polymers is comparable, it may because that the synthesis of compounds 1-4 used the same ligand(L1). Moreover, Compounds 1-4 have similar band gap values.

Comment 12: Figure 9. According this figure, an increase of catalyst loading above 10 mg does not allow to increase the reaction rate. The authors should continue these studies to show the optimal amount of catalyst. Please, complete these studies using 1, 3 and 5 mg of the catalyst, for example.

Reply: According to your suggestion, we conducted a thorough study under different catalyst dosage (5mg, 10mg, 20mg), and found that with the increase of catalyst dosage, the degradation rate increased first and then decreased, and the relevant description has been added to the original manuscript.

Comment 13: Heterogeneity of the reaction should be checked. Please, performed hot test.

Reply: Thanks for your suggestion. We believe that the heterogeneity of reaction is indeed an important exploration content, but we think that since we are focusing on the degradation of compounds 1-4 at room temperature, hot test may not be indispensable. We discussed this as a prospect for further research in the conclusion of the manuscript.

Comment 14: Please, compare the efficiency of your catalyst to that of classical photocatalyst Degussa P25 titanium dioxide, for example.

Reply: According to your suggestion, we compared the results with the reported photodegradation efficiency of Bi2Ti2O7/TiO2/RGO composite, and found that the degradation effect of compound 1 obtained by our one-step reaction is comparable with the modified Bi2Ti2O7/TiO2/RGO composite.

Comment 15: Elemental analyses do not correspond to propose structures. For example: for compound 1: Elemental Anal. Calc. for C19H23CuMo4N3O20 (772.54): C, 29.54; H, 290 2.98; N, 5.44. Found: C, 30.99; H, 3.01; N, 5.47.

Reply: Thanks for your reminder. we performed elemental analysis on compounds 1-4 again. We have revised it in original place and marked it yellow for you to view.

There are also minor points to correct:

Comment 1. The title of article is very long but contains only one keyword (ciprofloxacin) that is inconvenient for data bases. Please, change the title focusing on its research content.

Reply: Thanks for your valuable comments. The title of this article have been modified to include as many keywords as possible. We marked it yellow in our manuscript.

Comment 2. The Figure 1 is missing in my version of the article.

Reply: Figure 1 shows the structural formula of ciprofloxacin, we put it in introduction part.

Comment 3. Page 2, line 65 “a long time [18-20]”: Please add a point after [18-20].

Reply: Thanks for your reminder. We have added a point after [18-20] and marked it in yellow for you to view.

Comment 4. Page 3, line 102 “a porous 3D cation layer (Figure 2d)”. Please change this sentence. A layer possesses 2D structure and cannot be 3D.

Reply: Thanks for your reminder. We rewrote this sentence in the original manuscript and marked it in yellow for you to view.

Comment 5. Page 3, line 106 “forms a tetra coordination mode”. Please use standard way to describe coordination shell of metal ions.

Reply: We rewrote this sentence in the original manuscript and marked it in yellow for you to view.

Comment 6. Figure 2a and b. Use the same color code for Mo and Cu atoms.

Reply: Thanks for your reminder. We have modified Figure 2 according to your requirements and redrawn all crystal structure drawings to ensure the clarity of the picture.

Comment 7. Page 6, line 186. Please, use bold for labels 1-4.

Reply: Thanks for your reminder. We have modified the labels in its original place and marked it in yellow for you to view.

Comment 8. Please add points in journals abbreviation.

Reply: Thanks for your reminder. We have carefully checked the format of each reference to ensure that it meets the requirements.

Comment 9. Ref. 33. Please, delete [J]. and replace 13: by 13, (replace semicolon by comma)

Reply: Thanks for your reminder. We have modified it as required.

Reviewer 2 Report

Recommendation: Minor Revision

1)     DTG curve should also be provided along with TGA.

2)     The bandgap analysis could have been explained better. What cases the differences in the bandgap values of the compounds 1-4?

Author Response

                 Responses to the Reviewer # 2 s’ comments

Comment 1: DTG curve should also be provided along with TGA.

Reply: Thanks for your valuable suggestion. We have provided DTG curve along with TGA.

Comment 2: The bandgap analysis could have been explained better. What cases the differences in the bandgap values of the compounds 1-4?

Reply: We have added relevant descriptions according to your comments. For your convenience, we list the added contents below:

Bandgap is an important characteristic parameter of semiconductors. Its size related to the crystal structure and the bonding properties of atoms. The different bandgap values of compounds 1-4 maybe caused by their different crystal structure and the bonding properties. Based on the size of band gap and the reported literature, we speculated that the compounds 1-4 might have potential photocatalytic activity, so we carried out subsequent experimental exploration.

Reviewer 3 Report

An interesting knowledge has been proposed but however this manuscript required a major revision before acceptance

Comments

The novelty of the manuscript must be better emphasized 

The author must describe the importance of antibiotic degradation by MO 

need to be elaborated about the instrumentation facilities used in this manuscript in the Materials and methods section 

Add more discussion on TGA and band gap results

How do the size and shape effects photocatalytic property

update references: some of the references are too old. Updataion required, suggested to add very recent paper 

The authors should revise the conclusion part, add some more unique results and future prospectives 

Figure legends should contain more details 

Author Response

                     Responses to the Reviewer # 3 s’ comments

Comment 1: The novelty of the manuscript must be better emphasized

Reply: We added appropriate content in introduction to explain the novelty of the study.

Comment 2: The author must describe the importance of antibiotic degradation by MO

Reply: Thanks for your valuable comments on this manuscript. In introduction of this manuscript, we added examples of MO doped compounds that can efficiently degrade antibiotics. For your convenience, we list the added content in yellow below.

Comment 3: need to be elaborated about the instrumentation facilities used in this manuscript in the Materials and methods section

Reply: Thanks for your reminder. We have detailed the instruments and equipment used in this manuscript in the materials and methods section.

Comment 4: Add more discussion on TGA and band gap results

Reply: Thanks for your valuable suggestion. We have added more discussion on TGA and band gap results. We also have provided DTG curve along with TGA.

Comment 5: How do the size and shape effects photocatalytic property

Reply: Both size and shape are important factors affecting photocatalytic performance. When the size decreases, the band gap of the semiconductor becomes wider and the redox ability becomes stronger. At the same time, the smaller the size is, the shorter the time for the electron hole pair to reach the catalyst surface, and the faster it can react with the material adsorbed on the catalyst surface. However, the wider the band gap is, the weaker the light absorption capacity becomes. The shape determines the contact area between catalyst and reactant. Generally speaking, the larger the contact area, the better the catalytic performance. In our manuscript, we focused on the influence of external factors such as catalyst dosage, temperature and pH on the catalytic effect; Due to the recurrent pandemic, we cannot conduct the experiments about the influence of size and shape at present. We added this part to the conclusion as a prospect for future research.

Comment 6: update references: some of the references are too old. Updataion required, suggested to add very recent paper

Reply: Thanks for your valuable suggestion. We have updated the cited references and added several recently published references.

Comment 7: The authors should revise the conclusion part, add some more unique results and future prospectives.

Reply: We made appropriate modifications to the conclusion, deleted inappropriate expressions, and reorganized the description of this part. We also added some future prospectives.

Comment 8: Figure legends should contain more details.

Reply: We carefully checked the legend of each diagram and added detailed descriptions as much as possible. For example, we restated the legend of Fig. 3 and marked it with yellow.

Round 2

Reviewer 1 Report

Recommendation: The manuscript may be publishable after a major revision

Yunyin Niu and et al. carefully revisited and completed the article but I recommend this manuscript to publication only after corrections of following points (the first two points  were metioned on my first review)

1.     Please check the heterogenity of the reaction performing a filtration of the half of the reaction mixture at the moment when 50% of Ciprofloxacin is decomposed. Then, continue the irradiation of the reaction mixture and this filtrate to show that the decomposition does not observe in the filtered solution. This experiment is called “hot test” in catalysis. Based on the results of this experiment, we can conclude whether catalyst leaching in the solution influences on the catalytic reaction.

2.     Bi2Ti2O7/TiO2/RGO is a specific and not widely available material. It is difficult to evaluate the efficiency of new catalyst using this compound as a standard. Please, compare the efficiency of your catalysts to that of one of classical photocatalysts (Degussa P25 titanium dioxide, for example).

3.     All values of optical band gap were changed in revised manuscript as compared to the original version (Fig. 5) . Please, explain in detail in the Supporting Information how these values were obtained and meaning of red lines. The following reference can be useful: Solid State Communications 341 (2022) 114573.

There are also minor points to correct:

1.     I would recommend the authors to classify the prepared materials as coordination polymers because they are non-porous. MOFs generally are regarded as coordination networks with organic ligands containing potential voids. (see for example, Acc. Chem Res. 2005, 38, 273, Angew. Chem. Int. Ed. 2004, 43, 2334).

2.     Figure 3. Please, present the spectra according to alphabet order of letters ( (a) before (c)) and add in the caption the correspondence between the spectrum and the compounds (1- (a); ....)

3.     Please, complete the reference 13 by the number of manuscript (122121).

4.     Please, use only English in Scheme 1S.

5.     Figure S1, caption. Please, change D to minuscule in “Diffuse”.

6.     Synthesis 3 and 4. Please, Use M as an unite for the concentration given in mol/L.

Author Response

Response to the Reviewer#1’s comments

Comment 1: Please check the heterogeneity of the reaction performing a filtration of the half of the reaction mixture at the moment when 50% of Ciprofloxacin is decomposed. Then, continue the irradiation of the reaction mixture and this filtrate to show that the decomposition does not observe in the filtered solution. This experiment is called “hot test” in catalysis. Based on the results of this experiment, we can conclude whether catalyst leaching in the solution influences on the catalytic reaction.

Response: Thanks for your valuable suggestion. We have reviewed the relevant literature and prepared to conduct the “hot test” research of our materials according to your detailed introduction. Unfortunately, due to the severe situation of the COVID-19, we were quarantined and could not enter the laboratory for this study in a short time. According to the actual situation at present, we discussed this research in conclusion part as a prospect for further research.

Comment 2: Bi2Ti2O7/TiO2/RGO is a specific and not widely available material. It is difficult to evaluate the efficiency of new catalyst using this compound as a standard. Please, compare the efficiency of your catalysts to that of one of classical photocatalysts (Degussa P25 titanium dioxide, for example).

Response: Thanks for your valuable suggestion. According to your suggestion, we have prepared to compare the efficiency of our materials to Degussa P25 titanium dioxide. Also because of the COVID-19 epidemic, we cannot supplement this experiment in a short time. According to the actual situation at present, we can only compare our materials with the reported modified material (Bi2Ti2O7/TiO2/RGO).

Comment 3: All values of optical band gap were changed in revised manuscript as compared to the original version (Fig. 5). Please, explain in detail in the Supporting Information how these values were obtained and meaning of red lines. The following reference can be useful: Solid State Communications 341 (2022) 114573.

Response: According to the opinion of another reviewer, we provided the UV Vis diffuse reflection spectrum, recalculated the band gap value according to the method of the reported literature, and kept only two decimal places. The band gap energy (Eg) of compounds 1-4 was calculated according to Ahv = C(hv-Eg)2, where A is the absorption coefffcient, h is Planck constant, v is incident light frequency, and C is the constant. [Materials Chemistry and Physics 256 (2020) 123650]

There are also minor points to correct:

Comment 1: I would recommend the authors to classify the prepared materials as coordination polymers because they are non-porous. MOFs generally are regarded as coordination networks with organic ligands containing potential voids. (see for example, Acc. Chem Res. 2005, 38, 273, Angew. Chem. Int. Ed. 2004, 43, 2334).

Response: According to your suggestion, we have classified the prepared materials as coordination polymers, and revised the relevant contents of the title and introduction in the manuscript.

Comment 2: Figure 3. Please, present the spectra according to alphabet order of letters ( (a) before (c)) and add in the caption the correspondence between the spectrum and the compounds (1- (a); ....)

Response: Thanks for your reminder. We have modified it in its original place.

Comment 3: Please, complete the reference 13 by the number of manuscript (122121).

Response: Thanks for your reminder. We have modified it in its original place and marked it in yellow for you to view.

Comment 4: Please, use only English in Scheme 1S.

Response: Thanks for your reminder. We have modified it in its original place.

Comment 5: Figure S1, caption. Please, change D to minuscule in “Diffuse”.

Response: Thanks for your reminder. We have modified it in its original place and marked it in yellow for you to view.

Comment 6: Synthesis 3 and 4. Please, Use M as an unite for the concentration given in mol/L.

Response: Thanks for your reminder. We have modified it in its original place and marked it in yellow for you to view.

Reviewer 3 Report

ACCEPT

In revised form, this manuscript justified all comments and has improved significantly to justify a publication

Author Response

                      Responses to the reviewer 3 ’s comments 

Comments: ACCEPT

In revised form, this manuscript justified all comments and has improved significantly to justify a publication

 Responses: Thank you!